# Encapsulation of Rich-Carotenoids Extract from Guaraná (*Paullinia cupana*) Byproduct by a Combination of Spray Drying and Spray Chilling

**DOI:** 10.3390/foods11172557

**Published:** 2022-08-24

**Authors:** Lorena Silva Pinho, Priscilla Magalhães de Lima, Samuel Henrique Gomes de Sá, Da Chen, Osvaldo H. Campanella, Christianne Elisabete da Costa Rodrigues, Carmen Sílvia Favaro-Trindade

**Affiliations:** 1Departamento de Engenharia de Alimentos, Faculdade de Zootecnia e Engenharia de Alimentos, Universidade de São Paulo, Pirassununga 13635-900, São Paulo, Brazil; 2Department of Food Science and Technology, Ohio State University, Columbus, OH 43210, USA; 3Department of Animals, Veterinary and Food Sciences, University of Idaho, 875 Perimeter Drive, Moscow, ID 83844, USA

**Keywords:** microencapsulation, storage stability, carotenoid degradation, gum arabic, vegetable fat

## Abstract

Guaraná byproducts are rich in carotenoids, featuring strong antioxidant capacity and health-promoting benefits. However, these compounds are highly susceptible to oxidation and isomerization, which limits their applications in foods. This research aimed to encapsulate the carotenoid-rich extract from reddish guaraná peels by spray drying (SD), chilling (SC), and their combination (SDC) using gum arabic and vegetable fat as carriers. The carotenoid-rich extract was analyzed as a control, and the formulations were prepared with the following core–carrier ratios: SD20 (20:80), SD25 (25:75), SD33 (33:67), SC20 (20:80), SC30 (30:70), SC40 (40:60), SDC10 (10:90), and SDC20 (20:80). The physicochemical properties of the formed microparticles were characterized, and their storage stability was evaluated over 90 days. Water activity of microparticles formed during the SD process increased during storage, whereas those formed by SC and SDC processes showed no changes in water activity. The formed microparticles exhibited color variation and size increase over time. Carotenoid degradation of the microparticles was described by zero-order kinetics for most treatments. Considering the higher carotenoid content and its stability, the optimum formulation for each process was selected to further analysis. Scanning electron micrographs revealed the spherical shape and absence of cracks on the microparticle surface, as well as size heterogeneity. SD increased the stability to oxidation of the carotenoid-rich extract by at least 52-fold, SC by threefold, and SDC by 545-fold. Analysis of the thermophysical properties suggested that the carrier and the process of encapsulation influence the powder’s thermal resistance. Water sorption data of the SDC microparticles depended on the blend of the carrier agents used in the process. Carotenoid encapsulation via an innovative combination of spray drying and spray chilling processes offers technological benefits, which could be applied as a promising alternative to protect valuable bioactive compounds.

## 1. Introduction

The use of byproducts from agro-industrial processes has been growing over recent years as a strategy to valorize them and reduce environmental impact. Fruit and vegetable peels, seeds, and pomace are byproducts of their processing and show great potential as starting materials for the extraction of bioactive compounds [1]. One fairly representative fruit is guaraná (*Paullinia cupana*), a Brazilian plant native to the Amazon basin. In this region, the flowering of this plant occurs during the dry season, induced by water deficiency. The guaraná fruits become mature 2–3 months after flowering with a peel color of yellow to red [2].

Previous studies have found that guaraná seeds and peels are rich in alkaloids (caffeine, theobromine, and theophylline), polyphenols (catechin, epicatechin, and epicatechin gallate), and carotenoids (*β*-carotene and lutein) [3,4]. Carotenoids consist of natural pigments that are present in plants and other photosynthetic systems, and they are often used as food colorants. They also possess antioxidant capacity and have been claimed to act as anticancer agents, immune response stimulants, and pro-vitamin A activity promoters [5,6,7,8,9,10]. However, carotenoids are easily oxidized or isomerized in the presence of oxygen, light, and metals, as well as when exposed to heat during processing and storage, leading to products with reduced color and bioactivity [11]. Considering the potential applications of carotenoid-rich extract in foods, increased stabilization against processing conditions should be guaranteed, which can be fulfilled through microencapsulation.

Microencapsulation is a widespread approach to preserve the bioactivity of oxygen- and light-sensitive compounds, mask unpleasant tastes, and control release of valuable compounds [12]. A process commonly used for microencapsulation of carotenoids is spray drying [13,14,15]. This process turns a liquid feed material, which is composed of an ingredient (core) and a carrier agent, into a powder. As a consequence of the continuous operation of liquid atomization at high temperature and fast drying, the microparticles are formed. This increases the convenience of further processing and transportation in addition to preservation of active compounds. The efficiency of encapsulation in spray drying depends mainly on the types of carrier and core materials used in addition to the drying operating conditions [16]. Water-soluble polymers such as modified starches, whey protein, maltodextrin, and gum arabic are well-known and widely used carrier materials [17,18,19]. Gum arabic is a hydrocolloid extensively employed for the spray drying of materials containing bioactive compounds, due to its emulsification capacity, high solubility, low viscosity, nontoxicity, and ideal retention properties.

Spray chilling is an alternative process for stabilizing sensitive compounds such as carotenoids. Unlike spray drying that operates at high temperatures, the formation of lipid microparticles trapping bioactive compounds by spray chilling occurs at low temperatures. The lower processing temperature results in smaller degradation of these compounds [20,21,22]. However, the technique has relatively low encapsulation efficiency, and degradation of the lipid carrier may occur. Different types of materials, such as vegetable fat, fatty acids, and/or waxes are used as encapsulating agents to produce solid lipid microparticles by spray chilling. Due to its lower price and high availability, the use of vegetable fat is more appropriate than other lipids for this application [23,24].

The combination of spray drying and chilling offers the possibility of using the advantage of the single techniques and has been shown to better preserve bioactive components considering the presence of a double-wall coverage. For example, Arslan-Tontul and Erbas [25] observed higher gastric and thermal resistance of particles containing prebiotics produced by the combination of spray drying and chilling than those observed by processing individually. Fadini et al. [26] reported similar findings on the protection of functional oils when using combined technologies. However, whether this applies to carotenoids remains unknown.

The current study aimed to produce and compare microparticles containing a carotenoid-rich extract from guaraná peels produced by spray drying, chilling, and their combination. The microparticles were monitored during storage in terms of water activity, size, color parameters, and carotenoid stability. The physicochemical properties of microparticles, including morphology, water sorption, and thermal and oxidative stability of carotenoids, were measured. The influence of different carrier materials on the properties of the microparticles was also investigated.

## 2. Material and Methods

### 2.1. Materials

Guaraná fruits were provided by the Executive Commission of the Rural Economic Recuperation Plan in Cacao (Taperoá, Bahia, Brazil). Gum arabic (Acacia gum—Nexira, Brazil) was used as the carrier in the spray drying process. Microparticles obtained by spray chilling were produced using Al Home P54 vegetable fat with a melting point of 54 °C (Cargill, Itumbiara, Goiás, Brazil) as the carrier material. Microparticles obtained by the combination of spray drying and chilling processes were prepared using gum arabic as the first wall material and vegetable fat as the second coating. Ethanol (purity ≥ 99%) was purchased from Êxodo Científica (Sumaré, Brazil). Petroleum ether (30–70 fraction) and magnesium chloride (MgCl_2_·6H_2_O) were acquired from Synth (Diadema, Brazil).

### 2.2. Production of Carotenoid-Rich Extract from Guaraná Peels

Guaraná peels were dried in a convective oven (Marconi, MA035/1152) at 50 °C for 18 h and stored at −20 °C until further analysis. For carotenoid extraction, the peels were mixed with ethanol in a ratio of 1:10 (peel/solvent, *w*/*v*). According to Pinho et al. [4] ethanol presented ideal performance when extracting carotenoids from guaraná peels compared to other solvent systems tested (such as hexane and ethyl acetate). Furthermore, the use of ethanol as solvent was based on the fact that it is recognized as safe and is obtained from renewable sources.

The mixture was shaken using an orbital shaker (Orbital Shaker Marconi, MA420, Piracicaba, SP) for 4 h at 50 °C and centrifuged at 7168× *g* for 10 min [4]. Sunflower oil was added to the supernatant with a final concentration of 3% and thoroughly mixed to minimize carotenoid degradation (observed during preliminary experiments). A final concentration of 3% of the oil was selected after considering the liquid–liquid equilibrium for the system composed of sunflower oil and ethanol [27]. Afterward, the material was concentrated using a rotary evaporator (TE-211 Tecnal, Piracicaba, Brazil) at 48 ± 2 °C to 20% of the initial volume. The concentrate was named “carotenoid-rich extract”.

### 2.3. Production of Microparticles from the Carotenoid-Rich Extract

Carotenoid-rich extracts were mixed with the carrier materials at different proportions for the production of carotenoid-enriched microparticles by spray drying and chilling. Furthermore, microparticles obtained by the combined spray drying and chilling processes were prepared using spray-dried microparticles as the core with a layer of vegetable fat as a carrier material. Formulations in the processes are shown in Table 1.

#### 2.3.1. Microparticles Obtained by Spray Drying (SD)

An emulsion was prepared by mixing the carotenoid-rich extract and a gum arabic solution (20% *w*/*v*), using an Ultra-Turrax^®^ IKA T25 (Labotechnic, Staufen, Germany) at 11,200× *g* for 3 min. The emulsion was atomized according to Rocha, Fávaro-Trindade, and Grosso [28] with slight modifications in the drying temperature. A spray dryer (Model MSD 1.0, Labmaq do Brasil, Ribeirão Preto, Brazil) equipped with a spray nozzle of 1.2 mm and inlet air temperature of 140 °C, air speed of 2.5 m/s, feed flow of 10 mL/min, and air pressure of 823,759 N/m^2^ was used. During the drying procedure, the emulsion fed to the dryer was kept under magnetic stirring.

#### 2.3.2. Microparticles Obtained by Spray Chilling (SC)

Solid lipid microparticles were prepared by spray chilling [29]. Dispersions with different proportions of carotenoid-rich extract and high-melting-point vegetable fat were prepared using an Ultra-Turrax^®^ IKA T25 (Labotechnic, Staufen, Germany) at 11,200× *g* for 3 min and 64 °C. The mixture was atomized using the same spray dryer equipment (Model MSD 1.0, Labmaq do Brasil, Ribeirão Preto, Brazil) coupled to a spray nozzle of 1.2 mm and the following conditions: 1.0 kgf/cm^2^ air pressure, a feed flow of 40 mL/min, and a temperature of t 13 °C.

#### 2.3.3. Microparticles Obtained by the Combination of Spray Drying and Chilling (SDC)

The SD33 formulation was firstly prepared by spray drying as described previously (Section 2.3.1). SDC microparticles were then prepared by dispersing the SD33 microparticles into vegetable fat (*w*/*w*) at 11,200× *g* for 3 min and 64 °C. The atomization conditions used were the same as described previously for the spray chilling process.

### 2.4. Characterization of Microparticles and Storage Stability

The microparticles were placed in vials and kept in desiccators containing saturated magnesium chloride solution to create a storage environment of 33% ± 5% relative humidity (RH), at 25 ± 5 °C for 90 days. Water activity, color, mean diameter, and particle size were determined at the beginning (time 0) and the end (after 90 days) of storage. The total carotenoid content of the samples was evaluated every 15 days [30].

#### 2.4.1. Water Activity

The water activity (a_w_) of the microparticles was measured utilizing an Aqualab instrument (Series 3 TE—Decagon Devices, Pullman, WA, USA) at room temperature. Measurements were conducted in triplicate.

#### 2.4.2. Instrumental Color Analysis

The color parameters L (luminosity), a* (red–green), and b* (yellow–blue) of the samples were evaluated using a portable colorimeter (Mini Scan XE Plus—Hunterlab, Reston, VA, USA). Measurements were executed in triplicate.

#### 2.4.3. Particle Size and Distribution

Particle size distribution and the mean diameters of the particles were determined using laser diffraction (SaldI-201-V, Shimadzu, Kyoto, Japan). Pure ethanol was used as the dispersant of microparticles obtained by spray drying, while distilled water was used for the microparticles prepared by spray chilling and the combined process. Measurements were performed in triplicate.

#### 2.4.4. Determination of Total Carotenoid Content in the Nonencapsulated and Encapsulated Extract

The total carotenoid content was determined according to a spectrophotometric method [11], using a UV/Vis spectrophotometer (Genesys 10S Thermo Scientific, São Paulo/SP, Brasil) at 450 nm.

Carotenoids were extracted following Pelissari et al. [29] with slight modifications of the solvents used. Carotenoid-rich extracts and SD microparticles were dispersed in hexane, while SC and SDC microparticles were dispersed in petroleum ether. The mixtures were agitated for 1 min and kept in an ultrasound bath (USC-1400, Unique, Indaiatuba, Brazil) for 20 min. Distilled water was then added and agitated (Multi Reax, Heidolph Instruments, Schwabach, Germany) for 2 min. They were then centrifuged at 4930× *g* for 10 min, and hexane or petroleum ether-rich phases were transferred to cuvettes. The total carotenoid content was determined according to the following equation:(1)$$C=A\times V\times 10^{4}/Abs_{1cm}^{1\%}\times m,$$
where C (μg/g) is the total carotenoid content, A (nm) is the absorbance of the sample at 450 nm, V (Ml) is the final volume, and $Abs_{1cm}^{1\%}$ the extinction coefficient of *β*-carotene, which has a value of 2592 cm^−1^ in petroleum ether [11] and 2560 cm^−1^ in hexane [31]; m is the sample mass (g). The determinations were performed in triplicate.

#### 2.4.5. Encapsulation Efficiency (EE)

Encapsulation efficiency (EE) was calculated as the ratio between the total carotenoid content present in the microparticles $(C_{0}$) and the total carotenoid content in the feed material ($C_{FM}$) before drying, as indicated by Equation (2) [12,32]. Microparticle production was conducted in triplicate.
(2)$$EE\left(\%\right)=(C_{0}/C_{FM})\times 100.$$

#### 2.4.6. Kinetics of Carotenoid Degradation

The degradation of carotenoids during storage was described using kinetic models to quantitatively describe the reactions that occur in the system. The kinetics and the corresponding reaction rate constants *k*_0_ and *k*_1_ were evaluated following zero- and first-order kinetics described by Equations (3) and (4), for nonencapsulated and encapsulated carotenoid-rich extract. The half-life of the first-order reaction was calculated according to Equation (5) [33].
(3)$$C_{t}-C_{0}=-k_{0}\times t,$$
(4)$$-\ln C_{t}/C_{0}=k_{1}\cdot t,$$
(5)$$t_{1/2}=\ln2/k_{1},$$
where $C_{0}$ (μg/g) is the initial carotenoid content (time 0), $C_{t}$ (μg/g) is the content at time *t*, and *t* is the time (days). The reaction rate constants for the zero-order kinetics *k*_0_ (µg/(g·s)) and first-order kinetics *k*_1_ (1/s) were obtained from the slopes of linear plots of (*C_t_* − *C_t_*_,0_) vs. *t* and ln *(C_t_*/*C_t_*_,0_) vs. *t*, respectively. The kinetic analysis was carried out using the program Origin Pro 8.5.

#### 2.4.7. Carotenoid Retention (CR)

Carotenoid retention (%) was determined according to the following equation:(6)$$CR=\left(C_{90}/C_{0}\right)\times 100,$$
where $C_{90}$ (μg/g) is the carotenoid concentration after 90 days of storage, whereas $C_{0}$ (μg/g) is the initial carotenoid concentration in the microparticles.

### 2.5. Analysis of Selected Microparticles

To further analyze the characteristics of the microparticles, three formulations were selected. The samples SD33, SC40, and SDC20 were chosen mainly on the basis of the results of the total carotenoid content and the reasonable stability observed during storage. They were evaluated in terms of their morphology, oxidative stability, thermal properties, and water vapor sorption, as described below.

#### 2.5.1. Scanning Electron Microscopy (SEM)

SD, SC, and SDC microparticles were placed on a double-sided carbon adhesive tape (Ted Pella Inc., Redding, CA, USA); they were coated with gold and analyzed using SEM (Benchtop Microscope Hitachi TM 300, Tokyo, Japan). SEM images of the microparticles were captured at an accelerating voltage of 15 Kv.

#### 2.5.2. Oxidative Stability

The oxidative stability of powders was conducted according to De Leonardis and Macciola [34]. Microparticles (2.0 g) were heated from 25 °C to 120 °C under a 20 L/h airflow, using a Rancimat instrument (model 873, Metrohm, Herisau, Switzerland). The analyses were carried out in duplicate, and results were expressed as the oxidation induction time.

#### 2.5.3. Differential Scanning Calorimetry (DSC)

Thermal properties of microparticles were evaluated using differential scanning calorimetry (DSC 2500, TA Instruments, New Castle, Delaware, DE, USA). Samples were weighed (~5 mg) into aluminum pans, sealed, and placed in the DSC instrument. An empty pan was used as a reference. Samples were equilibrated at 25 °C for 2 min, ramped to 300 °C at a rate of 10 °C/min, kept at an isothermal condition (T = 300 °C) for 2 min, and then cooled to 25 °C at a rate of 30 °C/min [35]. Data collection and analysis were conducted using Trios software (TA Instruments). Triplicates were run for each sample, and the average results were shown.

#### 2.5.4. Dynamic Vapor Sorption (DVS)

The water vapor sorption of the microparticles was conducted using a Dynamic Vapor Sorption instrument (Surface Measurement Systems Ltd., Allentown, PA, USA). The airflow in the DVS was compressed nitrogen flowing at 200 Ml/min. The changes in sample mass at various relative humidities (between 0% and 90%) were recorded continuously at 25 °C using the DVS Analysis Macro V6.1 software. Each sample was run in duplicate.

### 2.6. Statistical Analysis

Data were examined by analysis of variance (ANOVA) and Tukey’s test, at the 5% level significance, using the statistical program SAS (Statistic Analysis System) version 9.2.

## 3. Results and Discussion

### 3.1. Characterization of Microparticles and Their Changes during Storage

#### 3.1.1. Water Activity (a_w_)

The water activities (a_w_) of microparticles produced by spray drying, chilling, and the combined process combination loaded with carotenoid-rich extract are shown in Table 2. SD microparticles had the lowest a_w_, probably due to larger water evaporation at the higher inlet temperature (140 °C) of the spray drying process. After storage, a_w_ increased due to the particle hygroscopicity (water uptake) as a consequence of the presence of gum arabic, which was used as the carrier material.

In this system, the molecular mobility increased, and the microparticles absorbed water until reaching an equilibrium condition, in an intermediate relative humidity range of 35–40% at 25 °C. This condition creates a favorable environment for binding water, which is influenced by the highly branched structure of gum arabic and its high water affinity [36]. The water activity of SD microparticles was below the minimum value of 0.60, which corresponds to a satisfactory condition to avoid microbial growth. It is worth mentioning that, even during storage, these samples exhibited great stability.

SC microparticles had a higher a_w_ (0.80–0.95) than SD and SDC microparticles (0.41–0.52), and they varied with formulations. These a_w_ values measured are in agreement with the a_w_ range reported by Labuza [37], from 0.3 to 1.0, which is associated with the susceptibility of high lipid oxidation at lower water activity conditions. Silva et al. [38] presented similar results, with water activities ranging from 0.91 to 0.98, for solid lipid microparticles loaded with probiotics produced by spray chilling.

#### 3.1.2. Color Parameters

Color coordinates L*, a*, and b* were examined during storage time, as parameters associated with the chemical stability of carotenoids encapsulated within SD, SC, and SDC microparticles (Table 2). Color parameter analyses were conducted considering that the carotenoid-rich microparticles could be used as colorants with bioactivity. SD20, SC20, and SDC10 microparticles showed higher lightness (L*), compared to the other treatments obtained through the same process. This was due to the higher content of gum arabic and/or vegetable fat, as well as the lower ratio between the concentration of extract and SD microparticles in their formulations. During the stability test, L* of the SD microparticles increased. This can be associated with carotenoid degradation, which leads to changes in the color intensity of the microparticles. McClements [39] reported that the increase in particle dimensions caused by aggregation and/or its morphology is likely to influence the light of the samples, affecting their L* value.

Regarding the parameter a* (redness), all the microparticles had positive values, which implies the subtle redness of all samples. A relatively superior magnitude of a* was observed for the SD33 microparticles, indicating more redness compared to others. The formulation was composed of a high proportion of extract, and the evaporation of the carrier solution during the spray drying led to a higher concentration of carotenoid within the microparticles.

Concerning the parameter b* (yellowness), all formulations displayed a noticeable decrease in this value during storage. Indeed, the intensity change of the yellow/orange color (Δb*) of the microparticles during storage followed the order: SC30 = SDC10 < SC20 = SDC20 < SD20 = SC40 < SD25 < SD33. These findings suggest that encapsulated carotenoids were more stable in lipid microparticles (SC and SDC) obtained by spray chilling and the combined process.

#### 3.1.3. Mean Diameter and Particle Size Distribution

The volume-weighted mean diameter (D_4,3_) and particle size distribution are shown in Table 2 and Figure 1, respectively. SD20 microparticles had a significantly larger mean diameter than SD25 and SD33 microparticles. This was mainly attributed to the sample formulation. SD20 feed material had the highest concentration of the carrier agent and increased the viscosity, resulting in larger spray-dried particles compared to the others from the same process.

During storage, the size of SD microparticles had a slight increase, which was likely due to agglomeration promoted by adhesion forces. The high water content likely led to the microparticles remaining bonded after collision, promoting the formation of aggregates. Microparticle agglomeration may promote powder application by reducing dust formation [29].

Samples obtained by spray chilling had average sizes that changed over 90 days. In the spray chilling process, operating conditions such as temperature, pressure, cooling air speed, feed flow, and spray nozzle diameter may affect particle size measured by the mean diameter. In addition, intrinsic parameters, such as the composition of lipid carrier, and the ratio of the feed matrix (i.e., bioactive ingredient:carrier) can affect the size of the microparticles [40].

SDC10 microparticles were larger than the SDC20 ones after being freshly prepared. The mean diameter of the microparticles increased continuously throughout storage (*p* < 0.05), especially the SDC20 formulation. Microparticles produced by spray chilling are classified as matrices, where the compound to be protected is entrapped by the volume of the particle. In the SDC microparticle case, the microparticle structure is composed of vegetable fat (carrier) and SD microparticles (core). The observed broad range of the diameter after storage of the SDC20 microparticles could be attributed to the composition of the microparticles, which may promote aggregation. Pelissari et al. [29] reported that the agglomeration of particles formed by hydrogenated and interesterified vegetable oils as carrier material may be attributed to the presence of melted triacylglycerols, which favors the adherence among lipid particles.

Overall, the size distribution of all formulations exhibited a unimodal distribution before and after storage at 25 °C. Microparticles prepared by SD had smaller sizes compared to those of the SC and SDC microparticles, even though the same drying equipment, nozzle type, and diameter were used for atomization of the feed. The smaller droplet size of the emulsion in the feed material and the operational parameters, such as high temperature and pressure, might facilitate solvent evaporation and shorten the particle coat formation during the atomization by spray drying. Conversely, in the spray chilling process, the heat transfer between the cold air and the molten feed material is able to quickly solidify the coating material, maintaining the original size of the formed microparticle.

Particle size is a physical property relevant to many food applications, and it is also related to the sensory attributes of foods in which powders are incorporated [41]. Hansen, Allan-Wojtas, Jin, and Paulson [42] suggested that powders with particle sizes less than 100 μm would not affect the sensory quality of the food. Thus, microparticles produced by spray drying, chilling, and their combination prepared in this study had suitable sizes for practical applications. However, from an applied perspective, SD microparticles had a more suitable size for food supplementation. In addition to the particle size, the composition of microparticles produced by the different methods would affect their applications. For example, SD microparticles could be used to disperse and protect carotenoids during the processing of aqueous products, while microparticles obtained by SC and SDC would be preferable for applications that require nonaqueous media.

### 3.2. Encapsulation Efficiency (EE)

Obtaining microparticles with the best application properties is one of the primary goals of encapsulation. Factors related to the feed flow rates, inlet air temperature, carrier material type, and formulation are essential because they affect the microparticles’ characteristics and encapsulation efficiency (EE). The EE of microparticles produced through spray drying, chilling, and the combined processes is shown in Table 3. The samples had high EE, ranging from 90% to 100% for SD, 90% to 97% for SC, and 82% to 94% for the SDC techniques, which demonstrates the greater entrapment of the carotenoid content in the microparticles for all treatments used.

A slight carotenoid loss was observed in SD samples, which might have occurred during the atomization step, in which these components possibly adhered to the hot drying chamber walls during processing. On the other hand, in SC treatments, the degradation of carotenoids may be attributed to the fact that the vegetable fat mixed with the extract was kept at 64 °C for a long time during the processing, to improve the mixture homogenization before atomization. Although spray chilling is a technique that produces microparticles at low temperatures, due to the fact that the fat used as the carrier needs to be in a molten state, a relatively high temperature must be used, which might affect the stability of carotenoids. Additionally, exposure of the product to environmental conditions, such as light and oxygen, may have triggered some oxidation of the pigments.

Concerning the SDC microparticles, they had lower EE compared to the others. Lipids exhibit three types of crystal polymorphic phases, i.e., alpha (*α*), beta-prime (*β*’), and beta (*β*), from the least to the most stable [43,44]. Thus, the carrier material can solidify following different crystallization patterns during the spray chilling process. Through atomization at low temperatures via rapid cooling, the solidification of microparticles composed of gum arabic and vegetable fat may generate an arrangement of lipid crystals that are unable to trap some of the bioactive compounds. As a result, the achieved arrangement probably led to the expulsion of carotenoids and their loss during production. A similar finding was reported by Navarro-Guajardo et al. [45].

### 3.3. Chemical Stability of Nonencapsulated and Encapsulated Carotenoid-Rich Extract

Carotenoid retentions of nonencapsulated extracts, as well as SD, SC, and SDC particles, upon storage were calculated using Equation (6) and are shown in Figure 2. It can be seen that carotenoids were degraded to a large extent during storage, despite reasonable storage conditions and the exclusion of light. Samples prepared by SD, SC, and SDC had retentions ranging from 97% to 68%, from 99% to 84%, and from 99% to 75%, respectively.

Throughout the 90 days of storage time, carotenoid losses increased, despite the encapsulation processes providing protection of total carotenoids when compared to nonencapsulated extracts. However, microparticles obtained by SC and SDC displayed higher effectiveness in preserving the carotenoids, which could indicate that the composition of the carriers impacted the stability of active components. This result was in agreement with the color change observed for the different samples. According to previous studies on guaraná byproducts, the major carotenoids found in guaraná peels are all-*trans*-*β*-carotene, followed by *cis*-*β*-carotenes and lutein. However, environmental factors such as soil condition, weather, and ripening process may influence the composition of guaraná peels [4].

The potential entrapment of oxygen within the carrier matrix in the process might contribute to the oxidation of carotenoids during storage. Furthermore, isomerization may also affect the stability of these compounds. Khoo, Prasad, Kong, Jiang, and Ismail [46] and Provesi, Dias, and Amante [47] claimed that light and temperature are the major causes of carotenoids isomerization. However, Rodriguez-Amaya [48] reported that degradation of carotenoids during storage is mainly caused by enzymatic and nonenzymatic oxidation, which correlates with the oxygen level in the microparticles and the carotenoid molecular structure.

The kinetics of the carotenoid degradation was investigated, and the results are presented in Table 4. Data were fitted to the zero- and first-order kinetics according to the carotenoid content during storage. Total carotenoid retention decreased remarkably, and the fitting of the degradation and its rate were analyzed by the correlation coefficient (*R^2^*) and the reaction k values, respectively. For the zero-order model, *R^2^* ranged from 0.610 to 0.957, and the *k*_0_ value ranged from 0.021 to 1.558 µg/(g·s), which was associated with suitable fitting for most samples. Reasonable values of *R^2^* (0.755–0.940) combined with lower *k*_1_ values (0.001–0.006 1/s) were achieved using first-order kinetics.

The structural characteristics of the microparticles led to different results for the kinetic parameters obtained from the formulations and processes (spray drying, chilling, and their combination). Therefore, according to these aspects, microparticles produced by spray chilling showed superior protective capacity, ensuring greater stability of the carotenoids over 90 days, as demonstrated by the half-life data.

It is worth noting that, when changing the carotenoid proportion in SD, SC, and SDC microparticles, the half-life was shorter in formulations with higher carotenoid content. This suggests that the bioactive components are less protected in such samples since more carotenoid molecules were likely located on the microparticles’ surface.

Taken together, the total carotenoid concentration present in the microparticles was considered as an indicator to assess the stability of microparticles during storage, how it was affected by the formation process, and the feasibility of using these samples for food applications. These results were used to design the next steps of the present research, in which further investigation may unravel more details about the design of suitable formulations for the different processes.

### 3.4. Characterization of Selected Microparticles by Morphology, Thermal Properties, and Moisture Sorption

#### 3.4.1. Scanning Electron Microscopy (SEM)

Scanning electron micrographs of SD33, SC40, and SDC20 microparticles loaded with carotenoids are illustrated in Figure 3. The SD33 microparticles exhibited different sizes with irregular spherical shapes, and no cracks were found at the outer surfaces, which is important to prevent gas permeability that may affect carotenoids protection. Similar behavior was observed in microparticles of eggplant peel extract [49] and pumpkin peel extract [50] encapsulated with gum arabic.

The cavities and irregularities on the microparticle surface obtained by spray drying are associated with rapid solvent evaporation and higher pressure inside the microparticles at high processing temperatures [51]. According to Elik, Yanık, and Göğüş [52], the temperature range between drying inlet air and the atomized droplets promotes wrinkling on the particle surface. This could be attributed to the initial expansion due to the incorporation of air in the particle, followed by its contraction in the colder chamber of the equipment. Bhusari, Muzaffar, and Kumar [53] proposed that the cavities in the microparticles containing gum arabic may be linked to the protein fraction in the gum that may promote the formation of small air pockets.

SC40 and SDC20 microparticles had spherical shapes of different sizes. The differences demonstrated that the addition of SD microparticles into the vegetable fat carrier may have affected their morphologies, mainly the surface of the microparticles. The SC40 powders showed some agglomeration and an irregular surface with some pores. Similarly, the solid lipid microparticles loaded with carotenoid [29] and guaraná seed extract [54] obtained by spray chilling showed a spherical shape with agglomeration. From a technological standpoint, these physical properties could reduce the flowability of the powders. Furthermore, holes on the microparticle surface can increase the exposure of carotenoids to oxygen, thus reducing the powder functionality in terms of its nutritional properties.

Regarding the SDC powders, the morphological analysis showed a smooth surface with a high level of agglomeration. This may be related to the complex carrier matrix formed by the combination of gum arabic and vegetable fat, as well as its water holding capacity. The collision of microparticles at moderate–high moisture contents might induce agglomeration; moreover, incomplete solidification of the carrier material could also lead to the formation of partially melted microparticles favoring agglomeration. The microparticle agglomeration can produce structures that no longer have a spherical shape. These structures may favor additional protection to the microencapsulated compounds, as the outer particles can shield the inner ones from environmental conditions. Indeed, microparticle morphology can be influenced by operational parameters, which include feed material composition, temperature, solvents used, and drying rate [55,56].

#### 3.4.2. Oxidative Stability

The Rancimat method is widely used to investigate the oxidative stability of oil/lipid-containing samples [57]. The technique is based on the measurements of an induction period (IP) linked to the formation of certain components when the lipid samples are oxidized under heating (at temperatures higher than 100 °C) and under a constant aeration flow (20 L/h). In the present study, a higher IP of the encapsulated extract compared to the nonencapsulated one led to a better stability of the carotenoids within the microparticles. Data for the nonencapsulated extract, which contained oil in its formulation, as well as for selected microparticles, are presented in Table 5. Results indicated that, under the conditions applied in the test, carotenoid-rich extracts were highly sensitive to oxidation. The oxidative stability improved significantly (*p* < 0.05) when microparticles were prepared using different carriers, which were assessed using a comparative approach. It was observed that spray drying encapsulation increased the thermal stability of the extract by at least 52-fold, in contrast spray chilling by threefold, and the combined process by 545-fold, confirming the ability of the encapsulated microparticles to protect sensitive compounds against degradation. SD encapsulation produced microparticles composed of a resistant shell, which had limited gas transfer. Regarding SC40 microparticles, although they had a lower IP compared to the other microparticles, they showed a high level of carotenoid retention after 90 days of storage (Figure 2). However, when the Rancimat test was used on the SC40 microparticles, the drastic treatment at high temperatures (120 °C) and high flow oxygen exposure caused morphological changes, increased surface area, and promoted the exposition of the lipid matrix and carotenoids, which led to rapid carotenoid degradation. When evaluating the effect of the microencapsulation combined techniques on oxidative stability, it was observed that the lack of porosity of the wall on the surfaces of SDC20 microparticles indicating a complete coverage was associated with the measured stability. This structure could have created a sturdier and dense shell matrix that reduced oxygen permeability into the microparticle core.

#### 3.4.3. Thermophysical Properties

DSC is a technique commonly used to evaluate the thermal properties of bioactive compounds inside matrices, detect melting points, and record variations in the crystal structure via the displacement or disappearance of the endothermic peaks in the analyzed material [58]. The DSC curves shown in Figure 4 represent the thermal profile of the different microparticles prepared by the different utilized encapsulation processes.

Regarding the first endothermic peak, for all formulations, the SD microparticles exhibited a transition that began at 141 °C and ended at 160 °C, with a melting temperature of 147 °C and an enthalpy of 8.3 ± 0.6 J/g powder. The broad curve was assumed to be related to the melting point of gum arabic, which agrees with the results of Mothé and Rao [59]. For the SC40 microparticles, the melting peak started at 39 °C and ended at 61 °C, having a T_m_ of 52 °C and enthalpy of 86 ± 2 J/g powder. For the SDC20 microparticles, the melting began at 38 °C and ended at 59 °C, with a T_m_ of 51 °C and enthalpy of 80 ± 2 J/g of. For both samples, SC40 and SDC20, the peaks described the melting range of the vegetable fat that was used as a carrier agent. Furthermore, among the three encapsulating processes used in this work, SD33, SC40, and SDC20 microparticles presented a second peak (highlighted in Figure 4), ranging from 177 to 200 °C. These peaks were probably associated with carotenoid melting at high temperatures. Similarly, in DSC curves of *β*-carotene, a large endothermic process was reported at about 180 °C [60]. Furthermore, Sy, Gleize, Dangles, Landrier, Veyrat, and Borel [61] found that the melting point of carotenoids ranged from 175 to 195 °C. According to the thermograms, the occurrence of two peaks suggested the formation of a heterogeneous system, and the changes in the thermal profile indicated interactions between the carrier and the core materials. The crystalline structure of microparticles suggested higher stability at room temperature and contributed to the protection of components against environmental factors [62]. In this system, the melting enthalpy was related to the energy absorbed to disrupt and breakdown the crystalline lipid structure of the carrier agent. Therefore, a more homogeneous crystalline structure necessitated a higher energy absorbed to carry out the melting transition. The melting points of the microparticles presented particular values, as different processes and carriers were used to produce the powders.

#### 3.4.4. Sorption Isotherms

Sorption isotherms usually express the relationship between the relative humidity (RH) surrounding the food and the equilibrium moisture content of the sample, at a certain temperature. These data are important in food processes and quality such as packaging and storage of materials to preserve the products. Figure 5 illustrates that SD33 samples exhibited significant changes throughout the whole range of RH used when compared to the other two samples. These microparticles were formulated with gum arabic as the carrier material, which has a composition of ~98 wt.% polysaccharides and ~2 wt.% proteinaceous material [63]. When exposed to atmospheric conditions of increasing RH, the hydrophilic domain of the polysaccharide tends to reach fast equilibrium moisture with the surrounding environment. Typically, atomized particles obtained by spray drying are hygroscopic and can easily absorb water [55].

The SC40 sample showed a lower increase in moisture content, even at a relative 95% RH. As expected, this sample presented minimum fluctuations in moisture, as the vegetable fat used for microencapsulation created a hydrophobic coating. SDC20, on the other hand, showed a slight increase in the moisture content up to 60% RH, and a large increase thereafter. The presence of gum arabic and vegetable fat in the microparticles possibly altered the balance of hydrophilic/hydrophobic interaction, favoring the two observed stages of water sorption at 25 °C.

By analyzing the sorption isotherms, it was possible to identify two zones: a region of slower adsorption at low and intermediate RH, and a region of capillary condensation, in which water absorption raised faster with increasing RH. Lourenço, Moldão-Martins, and Alves [56] reported similar results when studying the sorption isotherms of microparticles loaded with pineapple peel extract produced by spray drying using gum arabic as a carrier agent.

Generally, it can be seen that the behaviors of DVS curves were of type III (J shape). It is worth noting that, in the SD33 microparticles, this behavior was not that obvious. The isotherm profile by DVS analysis was a consequence of the physical and/or chemical transformations that occurred during the atomization, in addition to the carrier materials in the composition of the microparticles.

## 4. Conclusions

The production of microparticles loaded with carotenoids by spray drying, chilling, and a combined process was shown as an effective approach to provide pigment stability with a high encapsulation efficiency of 82% to 100%. The mean diameter and size distribution of microparticles exhibited an increase after 90 days; however, even after storage, they were within the suitable scale for industrial applications. In addition, all formulations exhibited variation in color over storage. The degradation kinetics of the carotenoids followed the zero-order kinetics for most samples, presumably driven by permeation of oxygen into the core of the microparticles. By using selected formulations for each process, micrographs demonstrated the physical details of the microparticles such as their shape, e.g., with an irregular surface for the SC microparticles and spherical shape for the microparticles prepared with the other two treatments. Micrographs also showed SD and SDC samples with a smooth surface and heterogeneous sizes. Although losses in the carotenoid content were noted in the SDC particles, they displayed good stability concerning their thermal properties, oxidative stability, and water sorption. Overall, the microencapsulation processes used in the present study not only provide practical options to improve carotenoid stability, but also explain the phenomenon behind pigment protection. Further research should be conducted to evaluate the carotenoid bioavailability within the different matrices obtained from the encapsulation techniques developed in this research for application in functional foods as colorants with bioactive potential.

## Figures and Tables

**Figure 1 foods-11-02557-f001:**
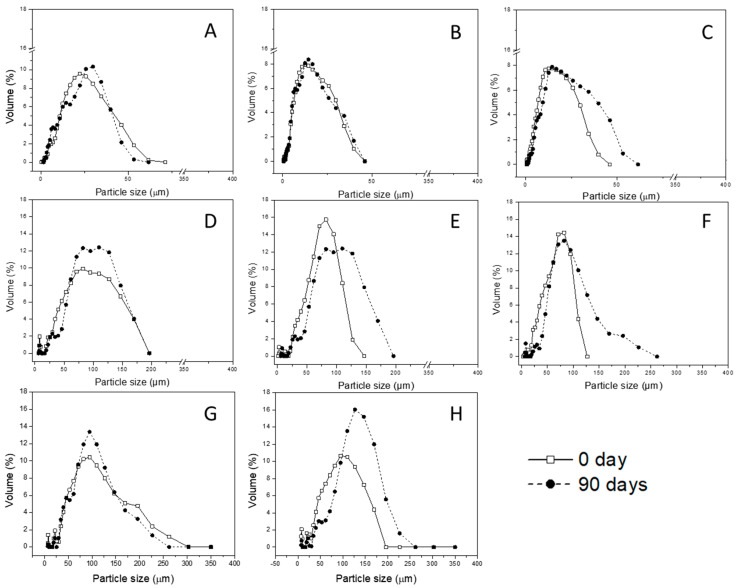
Particle size distribution of microparticles prepared by spray drying, chilling, and the combined process: (**A**–**C**) microparticles obtained by spray drying with a 20%, 25%, and 33% ratio core/carrier material, respectively; (**D**–**F**) microparticles obtained by spray chilling with a 20%, 30%, and 40% ratio core/carrier material, respectively; (**G**,**H**) microparticles obtained by spray drying and chilling combination with a 10% and 20% ratio core/carrier material, respectively.

**Figure 2 foods-11-02557-f002:**
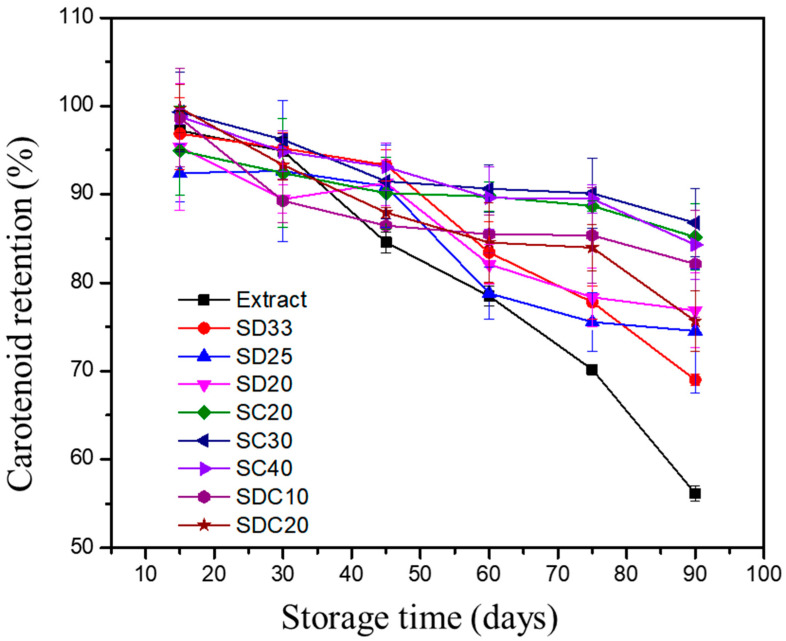
Carotenoid retention (CR, %) in nonencapsulated extracts and microparticles obtained by spray drying, chilling, and their combination, during 90 days of storage at 25 °C. SD: microparticles obtained by spray drying; SC: microparticles obtained by spray chilling; SDC: microparticles obtained by spray drying and chilling combination. The numerical suffix denotes the proportion of core/carrier material of each formulation.

**Figure 3 foods-11-02557-f003:**
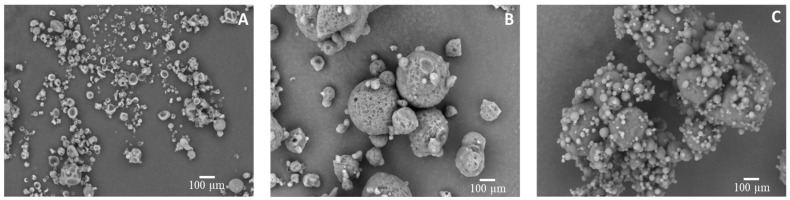
Scanning electron micrographs (1000× magnification) of microparticles obtained by (**A**) spray drying (SD33), (**B**) spray chilling (SC40), and (**C**) spray drying and chilling combination (SDC20). The numerical suffix denotes the core/wall material ratio of the selected formulation.

**Figure 4 foods-11-02557-f004:**
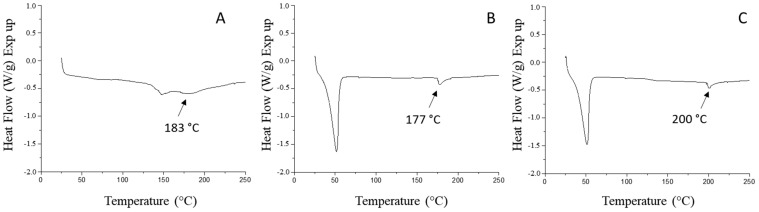
Differential scanning calorimetry (DSC) thermograms of microparticles prepared by (**A**) spray drying (SD), (**B**) spray chilling (SC), and (**C**) Spray drying and chilling combination (SDC). The numerical suffix denotes the core:wall material ratio of the selected formulation.

**Figure 5 foods-11-02557-f005:**
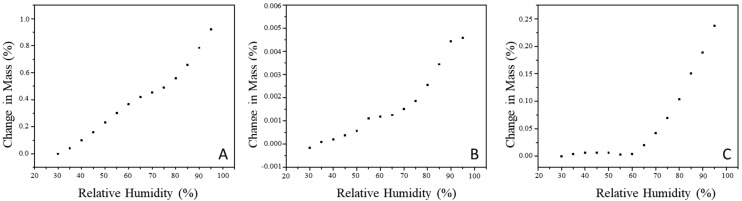
Sorption isotherms of microparticles obtained by (**A**) spray drying (SD33), (**B**) spray chilling (SC40), and (**C**) spray drying and chilling combination (SDC20). The numerical suffix denotes the core/carrier material ratio of the selected formulation.

**Table 1 foods-11-02557-t001:** Formulation and composition of microparticles produced by spray drying (SD), chilling (SC), and the combination of spray drying and chilling (SDC).

Formulation	Core (%)	Carrier Material (%)
	Extract	Gum Arabic solution (20%, *w*/*v*)
SD20	20	80
SD25	25	75
SD33	33	67
	Extract	Vegetable fat
SC20	20	80
SC30	30	70
SC40	40	60
	SD33 microparticles	Vegetable fat
SDC10	10	90
SDC20	20	80

**Table 2 foods-11-02557-t002:** Characterization of microparticles produced by spray drying (SD), chilling (SC), and their combination (SDC) before and after storage.

Response		Time (Days)	SD20	SD25	SD33	SC20	SC30	SC40	SDC10	SDC20
a_w_		0	0.132 ± 0.004 ^h,B^	0.210 ± 0.001 ^f,B^	0.161± 0.005 ^g,B^	0.913 ± 0.001 ^a,B^	0.872 ± 0.002 ^b,B^	0.801 ± 0.002 ^c,B^	0.47 ± 0.01 ^d,B^	0.412 ± 0.003 ^e,B^
		90	0.456 ± 0.002 ^e,A^	0.449 ± 0.001 ^e,A^	0.418 ± 0.003 ^e,A^	0.953 ± 0.004 ^a,A^	0.897 ± 0.006 ^b,A^	0.833 ± 0.008 ^c,A^	0.52 ± 0.01 ^d,A^	0.468 ± 0.008 ^e,A^
Color parameters	L*	0	80.61 ± 0.01 ^d,B^	76.44 ± 0.01 ^f,B^	74.11 ± 0.01 ^h,B^	84.15 ± 0.01 ^a,A^	82.51 ± 0.24 ^b,A^	79.20 ± 1.82 ^e,A^	81.26 ± 0.02 ^c,B^	74.39 ± 0.01 ^g,B^
		90	81.46 ± 0.01 ^b,A^	78.80 ± 0.01 ^c,A^	75.18 ± 0.01 ^f,A^	76.31 ± 0.01 ^d,B^	75.63 ± 1.42 ^e,B^	71.01 ± 2.29 ^h,B^	81.75 ± 0.01 ^a,A^	75.12 ± 0.01 ^g,A^
	a*	0	1.08 ± 0.04 ^d,A^	2.06 ± 0.03 ^b,A^	4.05 ± 0.02 ^a,A^	0.79 ± 0.02 ^e,B^	0.84 ± 0.26 ^e,B^	1.97 ± 016 ^b,B^	0.14 ± 0.02 ^f,A^	1.81 ± 0.05 ^c,A^
		90	0.51 ± 0.03 ^f,B^	1.04 ± 0.02 ^e,B^	3.17 ± 0.04 ^a,B^	1.08 ± 0.03 ^e,A^	1.42 ± 0.02 ^c,A^	2.29 ± 0.07 ^b,A^	0.06 ± 0.01 ^g,B^	1.60 ± 0.06 ^d,B^
	b*	0	23.07 ± 0.02 ^g,A^	33.39 ± 0.05 ^b,A^	39.68 ± 0.02 ^a,A^	27.29 ± 0.05 ^e,A^	24.12 ± 0.59 ^f,A^	30.76 ± 1.47 ^d,A^	20.35 ± 0.05 ^h,A^	31.04 ± 0.08 ^c,A^
		90	20.36 ± 0.05 ^g,B^	29.62 ± 0.02 ^b,B^	35.12 ± 0.07 ^a,B^	24.07 ± 0.09 ^e,B^	23.39 ± 0.03 ^f,B^	25.27 ± 0.07 ^d,B^	19.89 ± 0.06 ^h,B^	29.02 ± 0.10 ^c,B^
Mean diameter (µm)		0	16.2 ± 0.3 ^d,B^	10.8 ± 0.2 ^e,A^	10.9 ± 0.6 ^e,B^	59 ± 2 ^b,B^	56.6 ± 0.4 ^b,B^	52 ± 1 ^c,B^	73 ± 1 ^a,A^	60 ± 1 ^b,B^
		90	20 ± 1 ^c,A^	13 ± 2 ^c,A^	15 ± 2 ^c,A^	74 ± 2 ^b,A^	71.3 ± 0.8 ^b,A^	71.7 ± 0.8 ^b,A^	75 ± 2 ^b,A^	96 ± 2 ^a,A^

Values are the mean ± standard error (SE) (*n* = 3–4 analytical replicates). Different uppercase letters in a column represent a significant difference between 0 and 90 days with the response for formulations. The different lowercase letters in a row indicate a significant difference among formulations by the Tukey test at the 5% level of significance. SD: microparticles obtained by spray drying; SC: microparticles obtained by spray chilling; SDC: microparticles obtained by spray drying and chilling combination. The numerical suffix denotes the proportion of core/wall material of each formulation.

**Table 3 foods-11-02557-t003:** Encapsulation efficiency (%) of microparticles obtained by spray drying, chilling, and their combination.

Formulation	Encapsulation Efficiency (%)
SD20	96 ± 7 ^ab^
SD25	100 ± 2 ^a^
SD33	90 ± 2 ^bc^
SC20	96 ± 2 ^ab^
SC30	97 ± 1 ^ab^
SC40	90 ± 1 ^bc^
SDC10	82 ± 7 ^c^
SDC20	94.4 ± 0.5 ^ab^

SD: microparticles obtained by spray drying; SC: microparticles obtained by spray chilling; SDC: microparticles obtained by spray drying and chilling combination. The numerical suffix denotes the proportion of core/carrier material of each formulation. Different superscript letters indicate significant differences among the data in the same column (*p* < 0.05).

**Table 4 foods-11-02557-t004:** Rate constants (*k*_0_ and *k*_1_), coefficient of determination (*R^2^*), and half-life periods (*t*_1/2_) for degradation of nonencapsulated and encapsulated carotenoid-rich extracts obtained by different processes, during storage in the dark at 25 °C.

Sample	Zero-Order		First-Order		*t*_1/2_ (Days)
	*k*_0_ (µg/(g·s))	*R^2^*	*k*_1_ (1/s)	*R^2^*	
Nonencapsulated extract	1.558	0.957	0.006	0.857	108.08
SD20	0.089	0.911	0.003	0.901	236.93
SD25	0.176	0.844	0.003	0.847	212.20
SD33	0.284	0.805	0.004	0.783	168.07
SC20	0.025	0.925	0.001	0.919	606.36
SC30	0.035	0.909	0.001	0.792	581.55
SC40	0.075	0.949	0.001	0.859	546.95
SDC10	0.021	0.610	0.003	0.755	316.59
SDC20	0.058	0.942	0.003	0.940	223.45

SD: microparticles obtained by spray drying; SC: microparticles obtained by spray chilling; SDC: microparticles obtained by spray drying and chilling combination. The numerical suffix denotes the proportion of core/carrier material for each formulation.

**Table 5 foods-11-02557-t005:** Oxidation induction period of nonencapsulated and encapsulated carotenoid-rich extract, evaluated by Rancimat, at 120 °C.

Sample	Induction Time (h)
Nonencapsulated extract	0.04 ± 0.00 ^d^
SD33	2.10 ± 0.01 ^b^
SC40	0.11 ± 0.02 ^c^
SDC20	21.8 ± 0.3 ^a^

SD: microparticles obtained by spray drying; SC: microparticles obtained by spray chilling; SDC: microparticles obtained by spray drying and chilling combination. The numerical suffix denotes the core/wall material ratio of the selected formulation. Values with the same lowercase letter are not statistically different (*p* > 0.05).

## Data Availability

Data is contained within the article.

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
