# Peer review of "Encapsulation of Rich-Carotenoids Extract from Guaraná (Paullinia cupana) Byproduct by a Combination of Spray Drying and Spray Chilling"

_foods, 2022, doi:10.3390/foods11172557_

Round 1

Reviewer 1 Report

*** The whole paper is not prepared according to the Instruction for Authors! ***

There are additional comments for the paper:

1) The introduction is generic and on the student level. Please, rewrite this part as a scientific-relevant part with highlights in terms of novelty,  explanation of selected experimental design, etc.

2a) Growth and meteorological condition for selected fruit are required.

2b) In Results part also add potential variability of chemical composition based on different conditions during growth.

3) The discussion is very narrow and the results are insufficiently commented on and compared with the literature.

4) All Figures need to be improved;

Author Response

Reviewer #1: Comments and Suggestions for Authors
1) The introduction is generic and on the student level. Please, rewrite this part as a
scientific-relevant part with highlights in terms of novelty, explanation of selected
experimental design, etc.
Response: The introduction part has been revised accordingly to emphasize the novelty,
explanation of the experimental design as well as the scientific merits. For the detailed
changes, please see below:
a. More information on guaraná fruits has been added as “In this region, the flowering of
this plant occurs during the dry season, induced by water deficiency. The guaraná fruits
become mature 2-3 months after flowering with a peel color of yellow to red (Schimpl et
al., 2013).” L 62-65
b. Literature information has been added regarding the composition of guaraná seed and
peels. “Previous studies have found that guaraná seeds and peels are rich in alkaloids
(caffeine, theobromine, and theophylline), polyphenols (cat-echin, epicatechin, and
epicatechin gallate), and carotenoids (β -carotene and lutein) (Santana, Zanini, &
Macedo, 2020; Pinho et al., 2021).” L 66-69
c. The rationale of the current study has been added. “Considering the potential
applications of carotenoid-rich extract in foods, increased stabilization against
processing conditions should be guaranteed, which can be fulfilled through
microencapsulation.” L 80-83
d. The advantages of spray drying has been added. “This process turns a liquid feed
material, which is composed of an ingredient (core) and a carrier agent, into a powder.
As a consequence of the continuous operation of liquid atomization at high temperature
and fast drying, the microparticles are formed. This increases the convenience of further
processing and transportation in addition to preservation of active compounds.” L 90-96
e. The main task of the present work has also been added. “The physicochemical
properties of microparticles, including morphology, water sorption, and thermal and
oxidative stability of carotenoids, were measured. The influence of different carrier
materials on the properties of the microparticles was also investigated.” L134-138
2a) Growth and meteorological condition for selected fruit are required.
Response: This has been added to the introduction. Please see response above item a.
2b) In Results part also add potential variability of chemical composition based on
different conditions during growth.
Response: The effects of the potential variability of chemical composition were
addressed, as suggested by the Reviewer. This information was added as:
“According to previous studies on guaraná by-products, the major carotenoids found in
guaraná peels were all-trans-β-carotene, followed by cis-β-carotenes and lutein.
However, environmental factors such as soil condition, weather, and ripening process
may influence the composition of guaraná peels (Pinho et al., 2021).” L 496-500
3) The discussion is very narrow and the results are insufficiently commented on
and compared with the literature.
Response: More content regarding the description and comparison with literature has
been added to the discussion:
a. “This condition creates a favorable environment for binding water, which is influenced
by the highly branched structure of gum arabic and its high water affinity (Sanchez et al.,
2018).” L 341-344
b. “This was mainly attributed to the sample formulation. SD20 feed material had the
highest concentration of the carrier agent and increased the viscosity, resulting in larger
spray-dried particles compared to the others from the same process.” L 390-392
c. “Microparticle agglomeration may promote powder application by reducing dust formation
(Pelissari et al., 2016).” L 396-398
d. “Pelissari et al. (2016) reported that the agglomeration of particles formed by hydrogenated
and interesterified vegetable oils as carrier material may be attributed to the
presence of melted triacylglycerols, which favors the adherence among lipid particles.”
L 412-415
e. “Obtaining microparticles with the best application properties is one of the primary
goals for encapsulation. Factors related to the feed flow rates, inlet air temperature,
carrier materials type, and formulation are essential because they affect the
microparticle's characteristics and encapsulation efficiency (EE).” L 445-448
f. “It is worth noting that when changing the carotenoid proportion in SD, SC and SDC
microparticles, the half-life was shorter in formulations with higher carotenoid content.
This suggests that the bioactive components are less protected in such samples since more
carotenoid molecules were likely located on the microparticles' surface.” L 522-525
g. “Similar behavior was observed in microparticles of eggplant peel extract (Sarabandi
et al., 2019) and pumpkin peel extract (Lima et al., 2021) encapsulated with gum arabic.”
L 545-547
h. “Similarly, the solid lipid microparticles loaded with carotenoid (Pelissari et al., 2016)
and guaraná seed extract (Silva et al., 2019) obtained by spray chilling showed a
spherical shape with agglomeration.” L 567-569
i. “The microparticle’s agglomeration can produce structures that no longer have a
spherical shape. These structures may favor additional protection to the
microencapsulated compounds, as the outer particles can shield the inner ones from
environmental conditions. Indeed, microparticle morphology can be influenced by
operational parameters, which include feed material composition, temperature, solvents
used, and drying rate (Alves et al., 2017, Lourenço, Moldão-Martins, & Alves, 2020).” L
579-584
j. “Typically, atomized particles obtained by spray drying are hygroscopic and can easily
absorb water (Alves et al., 2017).” L 665-666
k. “By analyzing the sorption isotherms, it was possible to identify two zones: a region of
slower adsorption at low and intermediate RH, and a region of capillary condensation,
in which water absorption raised faster with increasing RH. Lourenço, Moldão-Martins,
& Alves (2020) reported similar results when studying the sorption isotherms of
microparticles loaded with pineapple peel extract produced by spray drying using gum
arabic as a carrier agent.” L 679-684
4) All Figures need to be improved.
Response: We had difficulty understanding the specific meaning of the reviewer. But we
made some changes to the figures so that they are easier to understand for the readers:
a. In Figures 1, 4 and 5 we made the font size larger on the x- and y-axis.
b. In Figure 2, the style of data plotting has been changed from column to line.
c. In Figure 3, we added a scale bar.

Reviewer 2 Report

The topic is interest to the field of food science and falls within the scope of the journal. However, the authors are recommended to carefully review the paper.

Following are few suggestions for the authors to improve the paper:

Kindly add a highlights section to the manuscript.

The Abstract part is clear and well aiming but needs minor adjustments related to the definition of the control and some percentages.

Introduction:

The Introduction part is clear and well structured. Also, all study's aims are easily understood and attained.

Material and Methods

1)          The Materials and methods part is well structured and aiming. It shows strength and innovative information related to the topic. All methods adopted were briefly and correctly explained.

2)          Carotenoids were extracted using polar solvent (ethanol), while the better for extraction using mixture of polar and non-polar solvent like: hexane:acetone:ethanol to establish the intensity of the electric field strength to obtain the highest extraction.

3)          What was the storage temperature for dehydrated samples? This is important because drying and chilling samples can sometimes agglomerate and adsorb moisture.

Results:

The Results and discussion part is a well-structured and aiming one. The scientific analysis of findings was appropriately performed. The statistical presentation of findings was correctly done. The findings were also well discussed based on previous related studies and literature. Some linguistic mistakes should be adjusted besides some sentences reformulation in a more appropriate manner.

1)          I suggest determination carotenoids by HPLC in guaraná peels extract to know the major carotenoids compounds

2)          Table 2: Mean diameter kindly write (µm) like figure 1.

3)          Fig. 2 is not clear. I suggest be a line not column

Conclusion part is well formulated and aiming. It summarizes correctly the main findings of the study. Also, authors suggested further and fruitful related research.

Author Response

Reviewer #2: Comments and Suggestions for Authors
The topic is interest to the field of food science and falls within the scope of the journal.
However, the authors are recommended to carefully review the paper.
Following are few suggestions for the authors to improve the paper:
Kindly add a highlights section to the manuscript.
Response: Highlights have been added to the text as follows:
• Combined spray drying and chilling produced stable carotenoid-rich particles.
• Encapsulated extracts exhibited higher thermal stability than those without
encapsulation.
• Gum arabic and hydrogenated fat were used as carriers to enhance the carotenoid
stability.
• The carriers used in each process tuned the particles’ physiochemical properties.
The Abstract part is clear and well aiming but needs minor adjustments related to the definition of the control and some percentages.
Response: Definition of the control and some percentages has been added to the text:
“gum arabic and hydrogenated vegetable fat as carriers. The carotenoid-rich ex-tract was analyzed as a control and the formulations were prepared following the core: carrier
ratios: SD20 (20:80), SD25 (25:75), SD33 (33:67), SC20 (20:80), SC30 (30:70),
SC40(40:60), SDC10(10:90), SDC20 (20:80).” L 31-34
Introduction:
The Introduction part is clear and well structured. Also, all study's aims are easily
understood and attained.
Response: Thanks for the supportive comments.
Material and Methods
1) The Materials and methods part is well structured and aiming. It shows strength and innovative information related to the topic. All methods adopted were briefly and correctly explained.
Response: Thank you for the comments. 2) Carotenoids were extracted using polar solvent (ethanol), while the better for extraction using mixture of polar and non-polar solvent like: hexane:acetone:ethanol to establish the intensity of the electric field strength to obtain the highest extraction. Response: Clarification of usage of ethanol as extracting solvent has been added to the method as “According to Pinho et al. (2021), ethanol presented ideal performance when extracting carotenoids from guaraná peels compared to other solvent systems tested (such as hexane, and ethyl acetate). Furthermore, the use of ethanol as solvent was based on the fact that it is recognized as safe and obtained from renewable sources.” L157-161 3) What was the storage temperature for dehydrated samples? This is important because drying and chilling samples can sometimes agglomerate and adsorb moisture. Response: The relevant information has been added to the method as: “The microparticles were placed in vials and kept in desiccators containing saturated magnesium chloride solution to create a storage environment of 33 ± 5 % relative humidity (RH), at 25 ± 5 °C for 90 days.” L 213-215
Results: The Results and discussion part is a well-structured and aiming one. The scientific analysis of findings was appropriately performed. The statistical presentation of findings was correctly done. The findings were also well discussed based on previous related studies and literature. Some linguistic mistakes should be adjusted besides some sentences reformulation in a more appropriate manner. 1) I suggest determination carotenoids by HPLC in guaraná peels extract to know the major carotenoids compounds Response Information on the major carotenoids in guaraná peeks has been added as: “According to previous studies on guaraná by-products, the major carotenoids found in guaraná peels were all-trans-β-carotene, followed by cis-β-carotenes and lutein. How-ever, environmental factors such as soil condition, weather, and ripening process may influence the composition of guaraná peels (Pinho et al., 2021). ” L 496-500
2) Table 2: Mean diameter kindly write (μm) like figure 1. Response: Thank you for the suggestion. The unit has been added. 3) Fig. 2 is not clear. I suggest be a line not column. Response: The style of data plotting has been changed from column to line to better display the changes in carotenoid retention with storage.

Conclusion part is well formulated and aiming. It summarizes correctly the main findings of the study. Also, authors suggested further and fruitful related research. Response: Thanks for the thoughtful comments.

Round 2

Reviewer 1 Report

Thank you for accepting all my suggestions.